# Quantitative Transcriptome Analysis of Purified Equine Mast Cells Identifies a Dominant Mucosal Mast Cell Population with Possible Inflammatory Functions in Airways of Asthmatic Horses

**DOI:** 10.3390/ijms232213976

**Published:** 2022-11-12

**Authors:** Srinivas Akula, Miia Riihimäki, Ida Waern, Magnus Åbrink, Amanda Raine, Lars Hellman, Sara Wernersson

**Affiliations:** 1Department of Anatomy, Physiology and Biochemistry, Swedish University of Agricultural Sciences, P.O. Box 7011, SE-750 07 Uppsala, Sweden; 2Department of Clinical Sciences, Faculty of Veterinary Medicine and Animal Science, Swedish University of Agricultural Sciences, SE-750 07 Uppsala, Sweden; 3Department of Biomedical Sciences and Veterinary Public Health, Swedish University of Agricultural Sciences, SE-750 07 Uppsala, Sweden; 4Science for Life Laboratory, Department of Medical Sciences, Uppsala University, SE-752 36 Uppsala, Sweden; 5Department of Cell and Molecular Biology, The Biomedical Center, Uppsala University, SE-751 24 Uppsala, Sweden

**Keywords:** mast cell, transcriptome, asthma, airway, horse

## Abstract

Asthma is a chronic inflammatory airway disease and a serious health problem in horses as well as in humans. In humans and mice, mast cells (MCs) are known to be directly involved in asthma pathology and subtypes of MCs accumulate in different lung and airway compartments. The role and phenotype of MCs in equine asthma has not been well documented, although an accumulation of MCs in bronchoalveolar lavage fluid (BALF) is frequently seen. To characterize the phenotype of airway MCs in equine asthma we here developed a protocol, based on MACS Tyto sorting, resulting in the isolation of 92.9% pure MCs from horse BALF. We then used quantitative transcriptome analyses to determine the gene expression profile of the purified MCs compared with total BALF cells. We found that the MCs exhibited a protease profile typical for the classical mucosal MC subtype, as demonstrated by the expression of tryptase (TPSB2) alone, with no expression of chymase (CMA1) or carboxypeptidase A3 (CPA3). Moreover, the expression of genes involved in antigen presentation and complement activation strongly implicates an inflammatory role for these MCs. This study provides a first insight into the phenotype of equine MCs in BALF and their potential role in the airways of asthmatic horses.

## 1. Introduction

Asthma is one of the most prevalent respiratory diseases in horses, characterized by chronic lower airway inflammation, mucus production and airway hyper-responsiveness [1]. Asthma in horses represents a major health issue with negative effects on animal welfare and physical performance. The disease is classified into mild/moderate asthma and severe asthma [1,2]. Mild/moderate asthma was formerly referred to as inflammatory airway disease (IAD) and is the most widespread form of asthma in horses, with poor performance as the main symptom and a reported prevalence of up to 80% in thoroughbred racing horses [3,4]. Severe asthma is affecting approximately 14% of horses in the Northern Hemisphere [5,6], and is characterized by recurrent airway obstruction resulting in breathing difficulties, frequent coughing and exercise intolerance. Due to high prevalence and its effects on animal welfare there is a need for increased understanding of the inflammatory mechanisms to improve management of equine asthma.

Airway inflammation in horses is primarily determined by bronchoalveolar lavage fluid (BALF) cytology [7]. An increased number of inflammatory cells in BALF such as >5% neutrophils, >2% mast cells (MCs) or >1% eosinophils, have been found to be more conclusive than case history alone in confirming an asthma diagnosis in horses [7,8,9]. However, the contribution of these inflammatory cell types to asthma pathology in horses is still elusive.

Mast cells are widespread tissue cells especially abundant in the skin and mucosal surfaces and mainly acting as sentinel cells of the immune system. Upon activation, mast cells release various inflammatory products, of which a majority are stored in secretory granules, including histamine, proteases and glycosaminoglycans [10,11]. The three major MC-restricted proteases are tryptase, chymase and carboxypeptidase A3 (CPA3), and the two forms of glycosaminoglycans in MCs granules are heparin and chondroitin sulfate. During activation, MCs also synthesize and release other types of inflammatory mediators such as cytokines and the arachidonic acid metabolites, leukotriene C4 (LTC4) and prostaglandin D2 (PGD2) [11,12].

It is now well established that MCs and MC products play a major role in asthma and this has been supported by a multitude of findings from both human and mouse [13,14,15]. For example, it has been shown that MCs accumulate in the airway smooth muscle, bronchial epithelia and alveolar parenchyma in human asthma and that MCs can correlate to asthma control, lung function and treatment response [16,17,18,19]. Moreover, a role for mast cells and their granule mediators has been supported by studies in mouse models of asthma [20,21,22]. Despite the increased understanding of the crucial role of MCs in human and experimental asthma, the involvement of MCs in horse asthma is less well documented. Interestingly though, an association between lung function and airway MCs has been demonstrated in equine asthma [7,23].

MCs are heterogeneous cells and they are traditionally classified into two main subtypes: the connective tissue type mast cell (CTMC); and the mucosal type mast cell (MMC) as originally described in rodents [24]. The majority of tissue resident MCs originates from an early wave of cells from the yolk sac, while MCs recruited upon inflammation originate from bone marrow-derived MCs precursors [25,26]. In humans, CTMCs contain both tryptase and chymase (and also CPA3) and are hence named MC_TC_, while the human counterpart of MMC, named MC_T_, contains only tryptase and no chymase or CPA3 [27]. Several findings in humans and mice have suggested a more complex diversity of MC phenotypes related to disease states and tissue environment [28]. For example, MCs with an unusual protease profile, expressing tryptase and CPA3, but not chymase, have been identified in the airway epithelium of patients with “Th2 high” asthma [17,29]. In support of an overall negative impact of these atypical MCs, tryptase is considered to contribute to asthma pathology whereas chymase may exhibit mainly protective functions, as demonstrated in mouse models [21,22]. A role for CPA3 in asthma has, however, not been conclusively demonstrated, although our recent findings suggested that CPA activity originating from CPA3 or other related carboxypeptidases could contribute to the pathology in experimental asthma [30].

MCs express a range of cell surface receptors, many of which are MC-specific or limited to a small number of cell types. The most important such receptors include FcεRI, which is the high-affinity receptor for IgE, CD117/c-Kit, which is the receptor for stem cell factor (SCF), and ST2, the receptor for interleukin-33 (IL-33) [31,32,33]. MCs also express the MAS-related G protein-coupled receptor X2 (MRGPRX2), which is a receptor for substance P and other positively charged low molecular weight ligands [34]. The expression level of surface receptors can differ between various MC subpopulations. For example, the MRGPRX2 receptor is expressed primarily in CTMCs although not in lung- or colon-CTMCs [28,35,36,37].

When it comes to lipid mediators there are also differences between CTMCs and MMCs. PGD2 is primarily produced by CTMCs, whereas MMCs produce high levels of LTC4 [37,38].

To summarize, the expression of granule proteases, as well as cell surface receptors and a variety of other proteins and mediators, can differ dramatically between different MC subtypes in humans and mice, including the classical CTMCs and MMCs, indicating that they exhibit partly different functions in tissue homeostasis, immunity and disease pathology. However, the phenotypes of equine MC subtypes in health and disease have not been described in detail.

To determine the type and characteristics of MCs in the airways of asthmatic horses we here developed a protocol to purify MCs from BALF and used a quantitative method to analyze their entire transcriptome. We show that these MCs are almost entirely of the MMC type and that the horse BALF is almost totally devoid of MCs of the CTMC type.

## 2. Results

### 2.1. Method to Isolate MCs from BALF of Asthmatic Horses

To study the phenotype of airway MCs in asthmatic horses we set out to optimize a method for isolation of horse MCs from BALF. Density gradient centrifugations can be used for isolation of rodent peritoneal MCs [39]; however, centrifugation of horse BALF cells using Percoll gradients of 40%, 60%, 70% or 90% did not enrich MCs. We then tested a number of commercially available antibodies for binding to horse MCs, most of which gave negative results or very weak binding (example of negative result in Figure 1A; tested antibodies are listed in Appendix A). However, we succeeded in isolating a population of MCs from horse BALF by using a mouse anti-horse IgE monoclonal antibody and the Miltenyi MACSQuant Tyto cell sorter. After sorting, a population of more than 90% IgE-positive cells was consistently obtained in three independent experiments with BALF from several horses, with representative results shown in Figure 1A, and with forward scatter (FSC) and side scatter shown in Appendix A. By using toluidine blue staining, the purity of the BALF MCs was estimated to be 92.9% (Figure 1B). Based on analyses of cell morphology, we found that the contaminating 7.1% non-MCs were mainly macrophages (Figure 1B), but we also found some lymphocytes and very few cells with a segmented nucleus. The isolated MCs and total BALF cells were, following RNA purification, used for transcriptome analysis.

### 2.2. Transcriptome Analysis of Purified MCs and Total BALF Cells

To characterize the phenotype of the purified BALF MCs we assessed the gene expression profile at a quantitative level using the Illumina-seq RNA-sequencing technology (provided by GENEWIZ, Leipzig, Germany). The transcriptome of total BALF cells from one sample was used as a reference. The data from the transcriptome analysis of three separate MC isolations (derived from two horses as described in Materials and Methods) were presented in the form of an Excel file including normalized TPM counts from over 22,000 different transcripts. From this large data set, we selected the most relevant groups of expression markers for deeper analyses and for direct comparison between MC-enriched and untouched BALF cells as presented in the sections below. Analyses of the highest expressed 100 genes demonstrated very similar expression levels between the three samples of isolated MCs (Appendix A). Data from one MC sample (termed MC1) was therefore selected as a representative sample in all further presentations as it was derived from the same sampling as the total BALF cell data, and hence allowed for direct comparison. A heat map with the differentially expressed genes in the isolated MCs versus the total BALF cells demonstrated several typical MC markers e.g., the FcεRI and tryptase as the top genes (Appendix A).

### 2.3. Protease Profiles Reveal a Typical Mucosal MC Population in Horse BALF

By focusing on the expression of the MC-specific granule proteases we could clearly see that the horse BALF MCs were of the mucosal type (MMCs), comparable in protease expression profile to MMCs found in humans. These MCs only expressed the beta-tryptase (TPSB2) with 1530 counts, and zero counts for both chymase (CMA1) and CPA3 (Table 1). The almost total absence of counts for both chymase and CPA3 in total BALF cells also indicated a very pure population of MMCs, and that the BALF contained very few CTMCs, if any.

When there are high amounts of proteases there is a need for control of their activity, which is to a large extent performed by protease inhibitors. We observed relatively high levels of a few protease inhibitors and several of these were expressed at much higher levels in the MCs compared to the total BALF cells (Table 1). The most highly expressed were cystatin C (CST3) with 6198 counts, SPINT2 with 244 counts, SERPINB9 with 237 counts and SERPINB6 with 230 counts (Table 1). Cystatin C is a potent inhibitor of lysosomal proteinases, primarily cysteine proteinases. SPINT2 is a membrane-bound inhibitor of serine proteases and SERPINB9 is an inhibitor of a number of extracellular serine proteases including members of the complement and coagulation systems.

### 2.4. Expression of FcεRI and Other MC Surface Markers

Another characteristic feature of MCs is their surface receptors, where the high-affinity IgE receptor, FcεRI, is one of the most important. The horse MCs were found to express high levels of FcεRI alpha chain (FCERIA) and gamma chain (FCERIG), but surprisingly low levels of the beta chain (MS4A2), with 1764, 3387 and 147 counts respectively (Table 2). They also showed relatively low levels of the SCF receptor CD117/c-kit and the IL-33 receptor ST2, with 164 and 58 counts respectively (Table 2). Notably, the BALF MCs did not express the MRGPRX2 receptor.

### 2.5. Low Expression of Toll-Like Receptors and Most Cytokine/Chemokine Receptors

MCs are sentinel cells, and as such they need to be able to respond to incoming pathogens and pathogen-specific molecules by pattern recognition receptors (PRRs). Some of the most well-known PRRs are the toll-like receptors (TLRs). We could only detect very low, or no, levels of transcripts for the TLRs, with 0–18 counts for TLR1-10 (Table 2). Low levels of expression were also detected for almost all cytokine and chemokine receptors with the only exception being the highly expressed CCR7 with 650 counts (Table 2). The other receptors with low but detectable levels were the IL-4 receptor (IL4R) with 78 counts, the IL-2 receptor beta chain (IL2RB) with 60 counts, the IL-13 receptor alpha chain (IL13RA1) with 54 counts, and the IL-10 receptor alpha chain (IL10RA) with 47 counts (Table 2). Interestingly, the histamine receptor 4 (HRH4) was expressed, with 24 counts, in the horse BALF MCs, which is similar to the low levels of this receptor that we found in mouse peritoneal MCs (Table 2) [35,36].

### 2.6. Enzymes Related to Histamine and Arachidonic Acid Metabolites

MCs are known to produce and store histamine and also to produce and release several arachidonic acid metabolites upon activation. To further characterize the horse BALF MCs, we therefore looked at the expression levels of enzymes involved in the synthesis of these low molecular weight inflammatory mediators. We found relatively low levels of the enzyme involved in the formation of histamine, the histidine decarboxylase (HDC, HS3ST1), with 166 counts (Table 3). In contrast, we found very high levels of the enzymes involved in LTC4 synthesis, including leukotriene C4 synthase (LTC4S), with 2413 counts. Enzymes involved in prostaglandin synthesis were expressed at significantly lower levels, which match previous findings concerning differences between MMCs and CTMCs [11,12]. Hematopoietic prostaglandin synthase, HPGDS, was the highest expressed with 441 counts, followed by prostaglandin-endoperoxide synthase 1 (PTGS1) with 297 counts (Table 3).

### 2.7. Genes Related to Glycosaminoglycans

Glycosaminoglycans, such as heparin or chondroitin sulfate, are important granule components of MCs. In mice, the highly sulfated heparin is the major glycosaminoglycan in CTMCs while the slightly less sulfated chondroitin sulfate dominates in MMCs. We here found that the horse MCs expressed high levels of the core protein for the synthesis of heparin and chondroitin sulfate, the serglycin (SRGN), with 5957 counts. However, there was a relatively low level of NDST-2, with 52 counts, which is the main enzyme responsible for sulfation of heparin (Table 3). Together these findings suggest that the horse BALF MCs primarily express chondroitin sulfate and no, or very low levels of, heparin. A list of the enzymes involved in chondroitin sulfate synthesis is also included in Table 3.

### 2.8. Expression of Transcription Factors

During cell differentiation, MCs express several transcription factors that are known to regulate this process. One such transcription factor, GATA2, is expressed at very high levels in both mouse peritoneal MCs and human skin MCs [35,36]. Interestingly, only a relatively low level of GATA2 was seen in the horse BALF MCs. However, GATA2 was found to be more abundant than both GATA1 and GATA3 in these cells, with 93, 59 and 20 counts respectively (Table 3). Both GATA1 and GATA2 are transcription factors known to be involved in the formation of MCs as well as in the transcriptional regulation of MC specific proteases such as the beta tryptase TPSB2, and of the IL-33 receptor ST2 [40,41].

### 2.9. Markers Related to Antigen Presentation and other Immune Related Functions

The high levels of some of the enzymes involved in leukotriene synthesis indicated that the horse BALF MCs could exert inflammatory functions. In line with this we could detect very high levels of several molecules involved in antigen presentation to T cells, including the MHC class II molecules. The horse MCs showed very high transcript levels for the MHC class II genes DRA, DQB, DRB, HLA-DMA, and CD74, with 17,906, 2988, 1990, 950 and 14,764 counts, respectively, as well as lower levels of DQA (78 counts) and HLA-DOB (120 counts) (Table 4). This finding suggests that lung MMCs may be involved in antigen presentation and thereby also actively drive the allergic T cell responses in asthmatic horses.

Furthermore, the presence of high transcript levels for the complement components C1QB and C1QC, with 1274 and 1525 counts, respectively, suggests that these MCs are also active players in the complement system (Table 4). Interestingly, also, was the relatively high transcript levels for several of the CD1 molecules in purified MCs compared to total BALF cells (Table 4). CD1 molecules are involved in non-classical antigen presentation and therefore in line with the high levels of MHC class II components, described above.

We also detected relatively high levels of transcripts for lysozyme, with 479 counts (Table 4). Lysozyme is an important enzyme for our defense against bacterial infections, by its cleavage activity against a bond between two sugar units in the bacterial cell wall, and thereby an important component of innate immunity.

With a few exceptions, the expression of most cytokines and chemokines was low (Table 5). The most highly expressed growth factor was TGF-β1 with 460 counts, followed by IL4L1 with 83 counts and IL-18 with 68 counts (Table 5). Several of the chemokines were expressed at significant levels including MIF with 631 counts, CCL24 with 268 counts, CCL8 with 242 counts, CCL5 with 163 counts and CCL22 with 59 counts (Table 5). However, the majority of the other cytokines and chemokines were expressed at very low levels (Table 5). Interestingly, there is also a low-level expression (46 counts) of leukemia inhibitory factor (LIF), a cytokine of importance for hematopoietic stem cells (Table 5).

A number of additional genes that were expressed at higher levels in the BALF MCs compared to the total BALF cells were identified. The expression levels for these genes are, together with members of the S100 family, the immunophilin family and a number of CD molecules, found in the Supplementary Appendix A. Some of them may be of major importance for the biology of the BALF MMCs.

## 3. Discussion

Asthma is one of the major health problems in horses, with significant negative consequences for animal welfare and physical performance. MCs are central players in both human asthma and in murine asthma models, however, their role in equine asthma has not been studied to the same extent. To address this issue, we here developed a protocol for isolating horse MCs from BALF of mild/moderately asthmatic horses and we analyzed the transcriptome of these MCs compared to that of total BALF cells. A comparison with results from previous studies of both mouse and human MCs showed major similarities but also a number of differences [35,36]. A summarizing scheme of the phenotype of the horse BALF MCs is shown in Figure 2. A key finding was that, out of the major granule proteases (tryptase, chymase and CPA3), only tryptase was expressed by the horse BALF MCs. This clearly demonstrates that these horse MCs are of the classical mucosal MC phenotype. Moreover, the almost total absence of transcripts for chymase and CPA3 in total BALF cells as well as in the isolated BALF MCs suggests that essentially all of the MCs in the BALF of these horses are of the MMC type (Table 1). Interestingly, a relatively recent study have shown that human lung MMCs express high levels of mRNA for both the tryptase and CPA3. However, the CPA3 mRNA in these MMCs does not seem to result in any production of the protein or alternatively that the protein is degraded in the lysosomal compartment due to lack of the chymase [42]. These human MCs are therefore only positive for tryptase and not CPA3. In this respect, the horse BALF MCs differed from their human counterpart as they were negative for CPA3 mRNA (Table 1).

In addition to the typical MMC protease profile expressed by the horse BALF MCs, i.e., tryptase, they also displayed several other features in support of an MMC phenotype. For example, the lack of substance P receptor MRGPRX2, the high level of leukotriene synthesis enzymes, the comparably low level of prostaglandin synthesis enzymes and the high levels of the major core protein for glycosaminoglycan synthesis in MC, serglycin, combined with a low level of the sulfotransferase enzyme NDST2, supports the idea that isolated BALF MCs have a mucosal phenotype.

The horse BALF MCs also expressed high levels of the high-affinity IgE receptor alpha and gamma chains but, surprisingly, very low levels of the beta chain. The expression level of the alpha chain was 12 times higher than the beta chain (Table 2). This indicates that the majority of the high-affinity IgE receptors on horse BALF MCs lack the beta chain. The expression of high-affinity IgE receptors consisting of only the alpha and gamma chains has also been shown for human dendritic cells [43]. An alternative scenario is that the lack of sufficient numbers of beta chains results in lower levels of the high-affinity IgE receptor on the surface of the horse BALF MCs. In addition to MCs, basophils also express high-affinity IgE receptors and could therefore potentially have been co-purified in the isolation procedure using anti-horse IgE antibodies. However, we did not find expression of the basophil markers MCPT-8, CLC and MBP in the isolated cells. Moreover, basophils are terminally differentiated cells and contain very small amounts of mRNA, which is why a potential basophil contamination would have a minimal effect on the transcriptome data, if any. Another disadvantage with using anti-IgE antibodies for purification is the potential risk of activating the MCs by cross-linking the FcεRI. To minimize this risk, we kept the cells on ice or at 4 °C during the entire isolation protocol. Based on the intact morphology of the isolated MCs (Figure 1B) we can exclude a general MC degranulation in response to the anti-IgE antibody.

Relatively low levels of the stem cell factor receptor, c-kit, and the IL-33 receptor, ST2, were detected in the horse BALF MCs, indicating a major difference from typical CTMCs concerning these receptors. Expression levels of both of these receptors can be at least one order of magnitude higher in both mouse and human CTMCs [35,36].

We also detected a major difference between the horse BALF MCs and the mouse and human CTMCs in the expression profile of transcription factors of the GATA-family. GATA-2 is very highly expressed in mouse peritoneal MCs in the range of several thousand counts, whereas we only found 93 counts for this transcription factor in the horse BALF MCs (Table 3) [35,36]. Both GATA1 and GATA-3 were also essentially absent in the mouse peritoneal MCs, compared to GATA-2, whereas in the horse BALF MCs the levels of GATA-1 and -3 were only slightly lower than that for GATA-2 (Table 3) [35]. However, the expression levels of all three transcription factors were relatively low in these horse MCs.

The detection of high levels of transcripts for the majority of the components of MHC class II as well as CD1 molecules indicate that the BALF MCs may participate in antigen presentation to T cells. The expression of high levels of both C1QB and C1QC, and lysozyme also indicate that they contribute to complement activation and in early innate responses to bacteria. It should be noted that macrophages in horse BALF, i.e., alveolar macrophages, could potentially express high levels of MHC class II, lysozyme and complement component C1q, which would also be in accordance with our recent study showing that mouse peritoneal macrophages and human peripheral blood monocytes express high levels of mRNA for both C1q and lysozyme [44,45]. It is therefore conceivable that a contamination of the isolated BALF MCs with substantial numbers of macrophages could have resulted in false detection of MHC class II, lysozyme and C1q in the present study. However, the horse MCs were 92.9% pure and even if all the remaining 7.1% of the cells were macrophages, their contribution to the mRNA expression would most likely not be sufficient to account for the high expression levels observed for these genes in the MC preparation. Despite alveolar macrophages being the dominating cell type in the horse BALF (65% of cells), the levels of MHC class II and lysozyme were in fact lower in the total BALF cells than in the isolated MCs, suggesting a relatively low expression of these genes in the alveolar macrophages of asthmatic horses. We also found low levels of MHC class II expression in mouse peritoneal macrophages [44]. In contrast, we have shown that both mouse macrophages and human monocytes express high levels of the two complement components, ficolin (FCN1) and properdin/complement factor P (CFP) [44,45]. Here we found high expression of ficolin and properdin also in the macrophage-abundant total BALF cells, but very low expression in the enriched MC population, hence arguing against a substantial contamination with macrophages. Collectively, these observations strongly suggest that MHC class II, CD1, C1q and lysozyme are indeed expressed by the horse BALF MCs.

Very low levels of the TLRs were observed in the horse MCs, indicating that they may have difficulties to act as sensitive sentinel cells. However, relatively low levels of PRRs, including TLRs, have also recently been observed in mouse peritoneal MCs and macrophages as well as in human monocytes, suggesting that relatively low levels of these receptors are still sufficient for a potent response to conserved pathogen-specific molecules [35,44,45].

One chemokine receptor, CCR7, was expressed at a high level, and a few others were expressed at low levels, i.e., CCR1, CCR2 and CCR5, in the horse BALF MCs (Table 2). CCR7 recognizes the chemokines CCL19 and 21 and may have a role in the migration of immune cells into the alveolar space.

We detected low levels of histamine receptor 4 (HRH4) and of leukemia inhibitory factor (LIF) in horse BALF MCs. Expression of these proteins have also been observed in mouse and human CTMCs but not in any of the other cell types (or tissue types) analyzed so far, indicating that this could be a common characteristic of MMCs and CTMCs in several mammalian species [35,36]. However, the functional significance of the selective HRH4 expression on MCs is still not known.

In conclusion, this study is the first in-depth analysis of horse MCs, demonstrating that a very pure population of MMCs can be isolated from the BALF of asthmatic horses. Interestingly, these cells displayed several inflammatory characteristics not previously identified in MMCs, such as the high level of MHC class II molecules, the classical complement component C1q and the antibacterial enzyme lysozyme. It is also possible that this phenotype is typical for horse MMCs under healthy conditions, demonstrating their normal function in the airways. However, we cannot exclude the fact that the BALF MCs analyzed in this study are a specific inflammatory type of MCs seen primarily in asthmatic lungs. To discriminate between these possibilities, and to address other questions concerning MCs in horses, we would need more detailed information of other MC populations in the horse.

Interestingly, a recent transcriptome study on horses with insect bite hypersensitivity found that several MC specific markers (FCERIA, MS4A2, CD117, and MRGPX2) were significantly upregulated in the skin, indicating the importance of MCs not only in asthma but also in inflammatory skin reactions in horses [46], where CTMCs are the dominating type of MCs. This study gives additional support for the importance of a better overall view of different horse MCs subtypes and their tissue distribution. This will allow a deeper understanding of the role of MCs in normal tissue homeostasis and during different inflammatory conditions in the horse. This could, for example, be performed by quantitative transcriptomic analyses of MCs purified from various tissues that are either healthy and diseased.

## 4. Materials and Methods

### 4.1. Horses and Collection of BALF

A total of three horse BALF samples were collected following a standard technique as described previously [47]. The cells of the BALF were pelleted by centrifugation at 400× *g* for 10 min at 4 °C and washed once in PBS with 10% FCS before use. Two of the BALF samples (BALF sample 1 and 2) were collected from the same horse having seasonal moderate equine asthma with mixed MC/eosinophil inflammation in the first BALF sampling, and normal cytology in the second sampling (after anthelmintic treatment and out of pollen season). The third sample (BALF sample 3) was collected from a horse with mild/moderate equine asthma with mixed MC/eosinophil inflammation, pituitary pars intermedia dysfunction (PPID) and moderate tracheal collapse. Cytology data for BALF sample 1 were 350 leukocytes (10 × 10^6^/L), 50% macrophages, 36% lymphocytes, 0% neutrophils, 4% MCs and 10% eosinophils; data for BALF sample 2 were 370 leukocytes (10 × 10^6^/L), 65% macrophages, 26% lymphocytes, 4% neutrophils, 3% MCs and 2% eosinophils; and data for BALF sample 3 were 250 leukocytes (10 × 10^6^/L), 56% macrophages, 28% lymphocytes, 3% neutrophils, 7% MCs and 6% eosinophils). Data derived from BALF sample 2, corresponding to the mast cell isolate denoted ‘MC1′, are shown unless otherwise stated.

### 4.2. Mast Cell Isolation

To isolate IgE-positive MCs, BALF cells were adjusted to 3.6 × 10^4^ cells/mL and were incubated for 30 min in 0.2 µg/mL mouse anti-horse IgE monoclonal antibody, clone c3h10 (Bio-Rad Laboratories, Solna, Sweden), and then incubated for 30 min with anti-mouse IgG1 monoclonal antibody, clone REA1017 Vio Bright B515, and anti-mouse CD117 antibody, clone REA791 APC (Miltenyi Biotec, Bergisch Gladbach, Germany). All washings and antibody incubations of the sample were performed in a PBS-based MACSQuant Tyto running buffer. The sample was placed into a MACSQuant Tyto cartridge and the IgE-positive cells were sorted by a MACSQuant Tyto cell sorter (Miltenyi Biotec, Bergisch Gladbach, Germany). The percentage of IgE-positive cells before and after sorting was analysed by a MACS Quant Analyzer 10. To avoid cell activation, the cells were kept on ice or at 4 °C during the whole procedure, until frozen.

### 4.3. Mast Cell Purity and Morphology

To assess the purity and morphology of the MCs, the sorted IgE-positive cells were cytospun onto glass slides using a Shandon Cytospin 2 (Thermo Fisher Scientific, Inc., Waltham, MA, USA) and allowed to dry before staining with toluidine blue (TB) using a standard protocol. An Olympus BX 60 microscope (40× magnification) was used to analyze and take photos of the cells. To calculate the purity of the isolated MCs, at least 200 cells per slide were counted and classified as either MCs (TB-stained) or non-MCs (TB-unstained).

### 4.4. RNA Purification and Quantitative Transcriptome Analysis

Total RNA was extracted from IgE-positive MCs and total BALF cells by using the RNeasy Plus mini kit (Qiagen, Hilden, Germany) according to the manufacturer’s protocol. The RNA was eluted with 30 μL of DEPC-treated water, and the RNA concentration was measured using a Nanodrop ND-1000 spectrophotometer (Nanodrop Technologies, Wilmington, DE, USA). The integrity of the RNA was confirmed by separation on a 1.2 percent agarose gel and visualization by ethidium bromide staining. Total RNA from the purified MCs and from total BALF cells was sent to GENEWIZ for transcriptome analysis by standard RNA-sequencing technology (GENEWIZ, Leipzig, Germany). In this process, the mRNA was fragmented, followed by polymerase chain reaction (PCR) and sequencing of 30 million fragments. Individual read lengths of 50 to 100 nucleotides were then compared to a transcriptome reference library (EquCab3.0). The transcript per million (TPM) values were normalized based on the total number of counts and the data were obtained as a large Excel file containing the total number of counts for each gene. Hence, the quantitative transcript levels of the most interesting MC-related genes could be analyzed.

### 4.5. RNA-Sequencing Data Analysis

The sequence reads were trimmed to remove possible adapter sequences and nucleotides with poor quality using Trimmomatic v.0.36. The trimmed reads were mapped to the Equus_caballus_ensembl reference genome available on ENSEMBL using the STAR aligner v.2.5.2b. The STAR aligner is a spliced aligner that detects splice junctions and incorporates them to help align the entire read sequences. The reads were mapped to the reference genome and unique gene hit counts calculated using feature counts from the Subread package v.1.5.2. Unique reads that fell within exon regions were counted.

### 4.6. Differential Gene Expression Analysis

After the extraction of gene hit counts, the gene hit counts table was used for downstream differential expression analysis. Using DESeq2, a comparison of gene expression between the total BALF cells and MCs. The Wald test was used to generate *p*-values and log2 fold changes. Genes with an adjusted *p*-value < 0.05 and absolute log2 fold change >1 were called differentially expressed genes for each comparison.

## Figures and Tables

**Figure 1 ijms-23-13976-f001:**
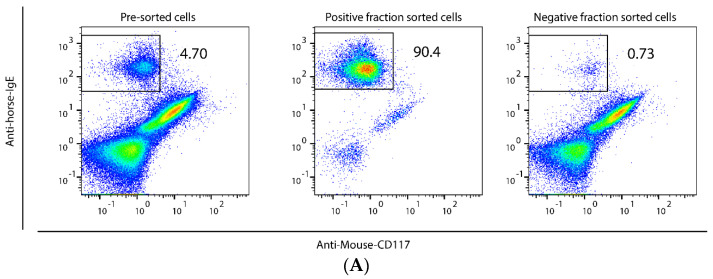
Mast cells (MCs) isolated from bronchoalveolar lavage fluid (BALF) of asthmatic horses. IgE-positive cells (MCs) were separated from total BALF cells by MACSQuant Tyto cell sorter using anti-horse IgE antibodies. A non-cross reacting anti-mouse CD117 antibody (clone REA791) was included as an example of a negative result in a screening of five potential MC-binding antibodies listed in Appendix A, and this antibody was also used to facilitate gating of cells. Panel (**A**) shows representative results from a total of three independent MC isolations, yielding 90.2%, 90.4% and 93.8% pure IgE-positive cells, respectively. The purity of the isolated MCs used in the transcriptome analysis was estimated to be 92.9% ± 0.12 (mean ± SEM) as determined by counting of Toluidine blue stained MCs (arrow) and non-stained non-MCs (*) for two samples (**B**).

**Figure 2 ijms-23-13976-f002:**
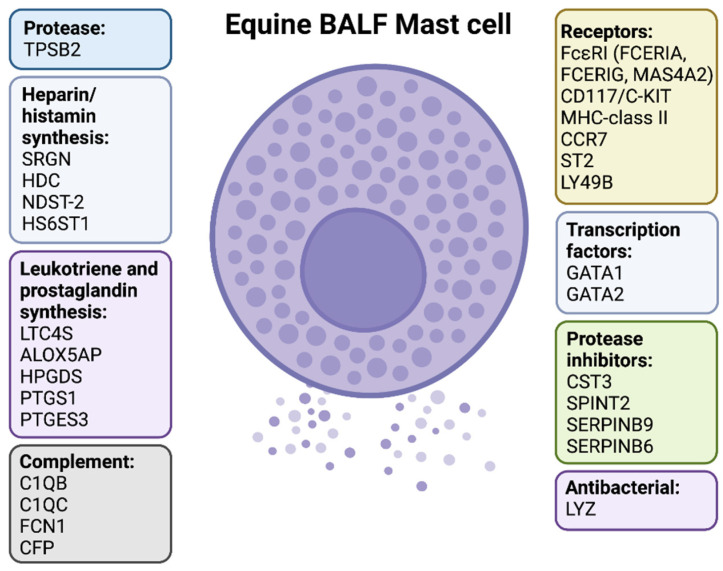
Summary of the phenotype of MCs isolated from BALF of asthmatic horses. Created with BioRender.com.

**Table 1 ijms-23-13976-t001:** Transcript levels for proteases and protease inhibitors in horse BALF MCs and total BALF cells. The number of normalized counts (TPM) was given for each gene (obtained from GENEWIZ).

	Genes	BALF MCs	Total BALF Cells
Proteases	CMA1	0	0.4
	CPA3	0	0
	TPSB2	1530	43
	TPSG1	16	0.3
	CTSG	0	0
	GZMA	68	166
	GZMK	63	109
	GZMM	2	2
	ELANE	0	0
	AZU1	0	0
	NSP4	0	0
	MMP2	82	1.3
	MMP9	28	3
	CTSC	275	100
	CPM	18	44
	CPD	11	5
	CPE	0	0
	CPZ	0	0
	CPA4	0	0
	CPA5	0	0
	CPA1	0	0
	CPB1	0	0
	CPA6	0	0
	CPB2	0	0
Protease inhibitors	CST3	6198	565
	SPINT2	244	16
	SERPINB9	237	18
	SERPINB6	230	27

**Table 2 ijms-23-13976-t002:** Transcript levels for immunoglobulin receptors, cytokine receptors, chemokine receptors, toll-like receptors and other receptors in horse BALF MCs and total BALF cells. The number of normalized counts (TPM) was given for each gene (obtained from GENEWIZ).

	Genes	BALF MCs	Total BALF Cells
Immunoglobulin Receptors	FCER1A	1764	43
	FCER1G	3387	4710
	MS4A2	147	1
	FCER2	5	0.6
	FCMR	2	0.8
Cytokine receptors	ST2	58	0.9
	C-KIT	164	1.4
	IL1R1	3	0.3
	IL1R2	5	0.6
	IL2RB	60	20
	IL4R	78	48
	IL9R	6	0.1
	IL10RA	47	28
	IL13RA1	54	18
	IL18R1	8	1
	IL21R	53	4
	IL27RA	11	77
Chemokine receptors	CCR1	35	1.9
	CCR2	21	14
	CCR3	5	6
	CCR5	13	5
	CCR7	650	26
Toll-like receptors	TLR1	18	4
	TLR2	16	47
	TLR3	13	11
	TLR4	0	0
	TLR5	1.2	0.2
	TLR6	8	5
	TRL7	17	16
	TLR8	13	32
	TLR9	2	0.4
	TLR10	13	0.2
Other receptors	LY49B	55	0.9
	MILR1	13	4
	HRH4	24	0.4

**Table 3 ijms-23-13976-t003:** Transcript levels for enzymes involved in the synthesis of heparin, histamine, chondroitin sulfate, leukotrienes, prostaglandins, and for a few transcription factors in horse BALF MCs and total BALF cells. The number of normalized counts (TPM) was given for each gene (obtained from GENEWIZ).

	Genes	BALF MCs	Total BALF Cells
Heparin and Histamine synthesis	HDC	166	2.5
	HS6ST1	69	8
	HS3ST1	47	2
	HS6ST2	2	0
	NDST2	52	4
	NDST1	1	3.5
	SRGN	5957	5935
Chondroitin sulfate synthesis	CSGALNACT1	1.2	2
	CSGALNACT2	21	7
	CHST3	0.1	0.6
	CHST7	45	5
	CHST11	1.5	10
	CHST13	32	2
	CHST15	1.5	0.4
	CHSY1	3	1.9
	CHSY3	0.03	2.5
	CHPF	34	24
Leukotriene synthesis	LTC4S	2413	109
	ALOX5	199	34
	ALOX5AP	2174	1887
Prostaglandin synthesis	HPGDS	441	16
	PTGS1	297	5
	PTGES3	216	211
	PTGDS	18	0.9
Transcription factors	GATA1	59	2
	GATA2	93	2.2
	GATA3	20	5

**Table 4 ijms-23-13976-t004:** Transcript levels for MHC class II, CD1, complement components and lysozyme in horse BALF MCs and total BALF cells. The number of normalized counts (TPM) was given for each gene (obtained from GENEWIZ).

	Genes	BALF MCs	Total BALF Cells
MHC-II and CD1	DRA	17,906	2344
	DQA	78	7
	DQB	2988	375
	DRB	1990	378
	HLA-DMA	950	460
	HLA-DOB	120	6
	CD74	14,764	3531
	CD1A3	426	44
	CD1A7	210	36
	CD1B1	157	5
	CD1B2	19	0.7
	CD1C	44	0.9
	CD1E1	195	6
	CD1E2	537	18
Complement components	C1QB	1274	3896
	C1QC	1525	6168
	FCN1	48	1480
	CFP	72	789
Lysozyme	LYZ	479	42

**Table 5 ijms-23-13976-t005:** Transcript levels for cytokines and chemokines in horse BALF MCs and total BALF cells. The number of normalized counts (TPM) was given for each gene (obtained from GENEWIZ).

	Genes	BALF MCs	Total BALF Cells
Cytokine and Chemokines	TGFB1	460	434
	IL1B	7	3
	IL4	1	0.4
	IL4L1	83	17
	IL5	6	0.4
	IL6	9	3
	IL13	1	0
	IL15	6	12
	IL17	9	3
	IL18	68	160
	IL25	0	0
	IL33	0	0
	LIF	46	1.3
	MIF	631	862
	CCL5	163	483
	CCL8	242	242
	CCL22	59	2
	CCL24	268	1211

## Data Availability

Data is contained within the article or Appendix A.

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
