# Peer review of "Quantitative Transcriptome Analysis of Purified Equine Mast Cells Identifies a Dominant Mucosal Mast Cell Population with Possible Inflammatory Functions in Airways of Asthmatic Horses"

_ijms, 2022, doi:10.3390/ijms232213976_

Round 1

Reviewer 1 Report

In this article, the authors reported the characteristics of the mast cell population involved in equine asthma. The authors collected BALF from horses with mild and moderate asthma symptoms and developed a novel protocol to recover mast cell populations in BALF with >90% purity.

The authors used this novel protocol to purify asthmatic horse

transcriptome analysis of mast cells. They report in detail the expression levels of mast cell-specific granule proteases, FceRI and receptors expressed on the cell surface, enzymes related to histamine and arachidonic acid metabolites, glycosaminoglycan-related genes, transcription factors, and markers of immune-related functions of antigen presentation and complement components in asthmatic horses.

Since the experimental materials were clinical specimen, BALF from asthmatic horses, the amounts of mast cells that can be collected were limited. However, this is a pioneering report that purified mast cell populations from BALF and analyzed their characteristics and phenotype. The authors used a mouse anti-human IgE monoclonal antibody to achieve isolation by MACS Tytosorting.

Data are presented in detail.

However, one point remains unclear.

1.     There should be 3 specimens (3 BALFs), however the transcriptome results in each table did not show S. E. Were the statistics processed?

A few minor comments are listed below.

2.     The scale bar should be present in the photo in Fig. 1b.

3.     A scheme summarizing the mast cell phenotypes (proteases, FceRI, etc.) of asthmatic horses revealed by the results of this paper would be helpful to better understand the results.

Reviewer 2 Report

1.Figure1. Why did you use CD117 here? if it is not important, you should show FSC or SSC. you mentioned the MCs was around 92%, how many samples did you do? and what are the else cells? Figure 1B is too simple, you should try other method to confirm MCs.

2.since you did RNAseq, how about your data? What are the top genes in the data? you should describe your sequencing data carefully, not just picked what you want to show.

3.How did you calculate TPM?

Round 2

Reviewer 2 Report

Authors have addressed my comments.